# How School Climate Affects the Development of the Social and Emotional Skills of Underprivileged-Background Students—An Empirical Study Based on the SSES2019 Data

**DOI:** 10.3390/children9121812

**Published:** 2022-11-24

**Authors:** Weihao Wang, Jian Xiao, Wenye Li, Jijun Yao

**Affiliations:** School of Education Science, Nanjing Normal University, Nanjing 210024, China

**Keywords:** underprivileged-background students, social and emotional skills, school climate

## Abstract

**Background:** Promoting the development of the social and emotional skills of underprivileged-background students is an integral part of educational and social equity. To date, there has been a lack of relevant research in this field. **Aims:** This study investigated the impacts of cooperative school climate and competitive school climate on the development of the social and emotional skills of underprivileged-background students. **Sample:** This study used the data of Chinese underprivileged-background students (N = 1739) from the Study on Social and Emotional Skills conducted by the Organisation for Economic Cooperation and Development (OECD-SSES2019). **Methods:** This study selected the ordinary least squares (OLS) analysis method and the quantile regression (QR) analysis method. **Results:** The ordinary least squares (OLS) analysis results showed that cooperative school climates promoted the development of various dimensions of the social and emotional skills of underprivileged-background students, while competitive school climates had significant negative impacts on the collaboration and emotional regulation of underprivileged-background students and had no significant impact on the other three major domains, namely engagement with others, open-mindedness, and task performance. A quantile regression analysis further explored the heterogeneity in the impacts of cooperative school climate and competitive school climate on the development of the social and emotional skills of underprivileged-background students through quantile regression and found that the impacts of a competitive school climate on underprivileged-background students with different levels of social and emotional skills were homogeneous, while the impacts of a competitive school climate on underprivileged-background students with different levels of social and emotional skills were heterogeneous. **Conclusions:** These findings provide a greater insight into the roles of school cooperative climate and school competitive climate in the development process of the social and emotional skills of underprivileged-background students.

## 1. Introduction

In recent years, social and emotional skills have drawn much attention from policymakers and educational researchers in various countries [1,2]. From an individual perspective, social and emotional skills, as key factors affecting personal success and well-being, are particularly important for the growth of adolescents and can, not only improve their relationships with peers and family members, but also reduce the incidence of bad behaviors such as drinking, truancy, and bullying [3,4,5]. From a social perspective, China’s exam-oriented education has long focused too much on students’ cognitive training and neglected their all-round development, inevitably resulting in problems such as “intellectual education emphasized, but moral education neglected” and “repetition emphasized, but innovation neglected”. In such a context, it is of vital importance to foster adolescents’ social and emotional skills.

Previous studies have shown that students’ families have an important influence on their growth [6,7,8] and that students from underprivileged family backgrounds tend to achieve poorer development than those from more advantaged family backgrounds [9,10,11]. Educational equity is a part of social equity, and educational equity in the modern sense should represent the separation of students’ achievements and their socio-economic background [12,13]. In China, the government attempts to ensure educational equity by providing students from underprivileged socio-economic backgrounds with more resources through reasonable allocation, thus counteracting the disadvantages brought by socio-economic factors for these students [14]. Eliminating the unequal impact of the socio-economic background of families on students’ development, through the efforts of social structures such as schools and communities, is an important way to achieve educational equity. The school climate is a key factor in school life and can regulate behaviors in school that are accepted by society, promote interaction among school members, and help them better develop in various ways [15]. In recent years, some researchers have found that a cooperative school climate and competitive school climate are closely related to students’ development [16,17]. However, due to the marginalized status of underprivileged-background students and the difficulty in measuring their social and emotional skills, few studies have been conducted to explore the impacts of cooperative school climate and competitive school climate on the development of the social and emotional skills of underprivileged-background students, based on large-scale empirical data. In such a context, this study investigated the impacts and heterogeneity of cooperative school climate and competitive school climate on the social and emotional skills of underprivileged-background students, based on data from the Survey on Social and Emotional Skills (SSES2019) Suzhou (People’s Republic of China).

## 2. Literature Review

The Coleman Report noted that families have a greater impact on students’ academic achievements than schools and communities [6]. Gershoff et al. (2007) found that the socio-economic backgrounds of adolescents’ families significantly affect the development of their social and emotional skills [18]. Students from low-income families tend to perform worse in terms of emotional regulation than those from high-income families, and they also face more severe behavioral problems [19]. Compared to students with high socio-economic status, those with low socio-economic status can achieve higher gains through the improvement of their social and emotional skills but tend to be outperformed by their counterparts with high socio-economic status in terms of these traits [20]. Liu’s study (2019) argued that strong social and emotional skills can regulate the effect of family socio-economic backgrounds on students’ academic achievements [21]. An experimental study on 33,000 sixth and seventh graders in Macedonia found that after a year of grit and self-regulation training, students from underprivileged backgrounds made more positive progress in social and emotional skills and academic achievements compared to students from average backgrounds [22]. This suggests that helping underprivileged-background students improve their social and emotional skills can alleviate the inequality in academic achievements caused by their families’ economic background, thereby promoting educational and social equity [11]. In recent years, some researchers have conducted studies on how to promote the development of the social and emotional skills of underprivileged-background students, and many believe that the adverse effects of an underprivileged socio-economic background on students can be moderated through the school climate [23,24,25].

Educational researchers in various countries have paid attention to school climate and realized its importance [26]. In this study, school climate refers to the relatively persistent and stable environmental characteristics of a school that are experienced by its members and influence their behavior [27,28]. Educational researchers and school leaders are increasingly interested in improving school climate, as they believe that an excellent plan to improve the school climate can make schools safer and more vibrant [29]. Students’ perceptions of school climate influence their academic, psychological, and behavioral development [30,31]. Positive and harmonious school climates improve academic performance and influence students’ physical and mental health, by reducing dangerous behaviors among students (e.g., fighting, smoking, etc.) and increasing positive behaviors (e.g., communication, cooperation, and interpersonal relationships among students) [32]. Liu and Lu (2012) conducted a study on 368 high school students in China and found that perceptions of school climate moderated the relationship between academic stress and the depressive symptoms of high school students [33]. Similarly, Luengo-Kanacri et al. (2017) conducted a longitudinal study on 151 adolescents and found that adolescents showed higher levels of pro-social behavior when they perceived a positive school climate. A positive school climate refers to an environment in school in which teachers and students care about and support each other, have consistent values, standards, goals, and the same sense of belonging, and participate in and influence the school’s organization and decision making [34]. Wong et al. (2021) argued that improving the school climate could be a strategy to improve students’ task performance and self-efficacy [35]. Therefore, a more positive school climate not only enables students to achieve better academic performance, feel happier, and have better physical and mental health, but also reduces the incidence of bad behaviors [36,37]. However, most previous studies have generally divided school climates into positive or negative [38], and only a few studies have discussed the impacts of school climate on student development from the perspectives of cooperation and competition [16,17]. As many schools promote students’ academic performance by creating a competitive climate, research on the impacts of competitive and cooperative climates may be of more practical value. Therefore, this study investigated the impacts of cooperative school climate and competitive school climate on the social and emotional skills of underprivileged-background students and focused on a heterogeneity analysis of the two types of school climate.

Competition and cooperation are two basic forms of human interaction and are ubiquitous phenomena in modern school environments, and all students have to master the ability to ensure all-round development in both school climates [39,40].

Darwinism argues that living creatures cannot evolve useful physiological structures and cognitive abilities without competition [41]. Modern education is characterized by a large scale. Large numbers of students gather, while the scarcity of educational resources leads to the fact that only a few students have access to high-quality educational resources, which justifies the existence and development of competition in education. Therefore, competition, as a part of the basic logic of modern education, should be included in educational research [42,43]. Taking China as an example, after the reform and opening up, people gradually realized the urgency of breaking the egalitarianism in education under the planned economy, and many scholars proposed introducing the concept of competition in the commodity economy into the field of education and to use education background as an appealing “reward”, to attract more people to compete for a higher status [44,45]. Wang et al. (2015) found, in a survey on the well-being of students serving as class representatives, that the stress created by a certain level of competition can significantly increase well-being [46]. Likewise, a study by Worm and Buch (2014) noted that students who engage in controlled competition have higher academic motivation and better grades [47]. Educational competition is, to some extent, a product and a sign of social change and one of the driving forces of educational development [45]. However, as competition intensifies in China’s society, competitive activities in education have become increasingly intense, and the idea of competition has been strengthened. As a result, competitive activities in education see a trend of abnormal development, which has a serious negative impact on educational practice [48]. Competition in schools is generally divided into self-competition, individual competition, and team competition. Self-competition emphasizes students’ self-comparative process. Individual competition refers to intra-group competition, and team competition focuses on intra-group cooperation and inter-group competition [49]. Excessive individual competition tends to cause stress and frustration among students with poor academic achievements and low self-efficacy, thus making them less confident and active in learning, while exam-oriented education is a product of uncontrolled individual competition [50,51]. Obviously, natural, fair, and controlled competition can make schools more vibrant and improve economic and social benefits, but the excessive competition caused by a one-sided competition-oriented approach is detrimental [52]. Therefore, it is crucial to control the direction and degree of competition in school education.

In real life, while competition is important, cooperation is often a necessary path to success [53]. In studies on schools, cooperation and competition are frequently discussed together. Katz et al. (2021) concluded that teamwork can improve productivity, promote team members’ learning and development, and reduce psychological problems [54]. Johnson et al. (2012) conducted a meta-analysis with 164 studies as samples and found that cooperative learning is more effective than competitive and individual learning [55]. Cooperative learning, not only enables students to acquire more knowledge, but also promotes the development of multi-dimensional social and emotional skills, including the willingness to share, help others, respect others, admit mistakes, and accept others’ opinions. In addition, it helps to reduce behaviors detrimental to society [55,56]. Li’s study (2021) based on PISA 2018 found that a highly cooperative school climate is more conducive to students’ development than a highly competitive one, while a highly cooperative school climate can also enable efficient division of labor and cooperation [16]. Wang and Wang (2022) found that a highly cooperative school climate promotes students’ development, in terms of non-cognitive abilities, while a highly competitive one negatively affects students’ mental health [17]. An experimental study by Llorent et al. (2022), on 156 students in grades four to six, found that learning through cooperation-based programs significantly improved students’ social and emotional skills [57], which suggests that a good cooperative school climate can promote the healthy and all-round development of students.

At present, schools place too much emphasis on the results of standardized tests, while ignoring the normal emotional needs of students. The school climates created by schools that consider academic performance a priority inevitably foster individualism and focus on competition, while ignoring cooperation. As a result, exclusion, alienation, and polarization tend to be found among students [58]. In the context of the high-quality development of education, it is more important to correctly understand the meaning and operating mechanisms of a competitive school climate and cooperative school climate, define the boundary between competition and cooperation in schools, and re-examine the relationship between competition and cooperation in education [16,52]. We reviewed the existing studies and found the following limitations: First, the existing studies mostly explored factors affecting students’ social and emotional skills from the perspective of individual characteristics, and few studies discussed the factors and measures affecting the development of the social and emotional skills of underprivileged-background students from the perspective of school climate. Second, the existing studies are mostly theoretical research, with small samples, and focused on negative mental health outcomes, with a lack of empirical studies based on large-scale surveys and focused on the positive mental health development of adolescents. Third, the existing studies mostly used multiple linear regression methods, aiming to explore the average effect of the impacts of competitive school climate and cooperative school climate on students’ development. The lack of heterogeneous discussion made it difficult for these studies to describe the impact of school climate in detail. In 2019, the Organisation for Economic Cooperation and Development (OECD) conducted a worldwide study on the social and emotional skills of adolescents (SSES). A framework for assessing the social and emotional skills of adolescents was established, based on the five-factor model of non-cognitive abilities for the study on adolescent students, in ten cities in nine countries around the world. The survey provided the data for this paper, and the data of underprivileged-background students in SSES2019 Suzhou (People’s Republic of China) were selected for this study. Such data can effectively guarantee the scientific validity and authority of the research [59]. At the same time, this study adopted a quantile regression method, in addition to multiple linear regression, to investigate the average effect and heterogeneity of the impacts of different levels of competitive school climate and cooperative school climate on the development of the social and emotional skills of underprivileged-background students. This study can, not only improve the application of cooperation–competition theory in the field of education, but also explore the impacts of competitive school climate and cooperative school climate on the development of the social and emotional skills of underprivileged-background students.

## 3. Data and Methods

### 3.1. Data

This study selected Chinese data in SSES2019. The OECD selected 3800 10-year-old and 3750 15-year-old students from 151 schools in six districts and four county-level cities under the jurisdiction of Suzhou City for the survey [60]. As this study mainly investigated the factors affecting the development of the social and emotional skills of underprivileged-background students in China, students from the poorest 25% of socio-economic backgrounds were selected as underprivileged-background students. A total of 1739 valid samples were obtained after data sorting, matching, and cleaning, with invalid samples being eliminated.

### 3.2. Variable Description

The dependent variables in this study are the scores of five domains of social and emotional skills: collaboration (COL), emotion regulation (EMO), engagement with others (ENG), open-mindedness (OPE), and task performance (TAS). According to the OECD (2018) classification [59], this study used the mean value of the scores of cooperation, trust, and empathy as the score of COL; the mean value of the scores of emotional control, optimism, and stress resistance as the score of EMO; the mean value of the scores of sociability, assertiveness, and energy as the score of ENG; the mean value of the scores of tolerance, curiosity, and creativity as the score of OPE; and the mean value of the scores of self-control, responsibility, and persistence as the score of TAS.

The core independent variables in this study are cooperative school climate and competitive school climate. Two items in the questionnaire (a. in school, students attach importance to cooperation, such as co-learning; b. in school, students are cooperating) measured the cooperative school climate, and another two items (c. in school, students attach importance to competition, such as mutual competition; d. in school, students are in competing) measured the competitive school climate. The options ranged from 1 to 4, representing never, sometimes, often, and always, respectively. In the multiple regression analysis, the mean value of items a and b and that of items c and d were used as the scores of the cooperative school climate and the competitive school climate, respectively.

This study referred to previous studies and included gender, age, being an only child, preschool education experience, and physical health as control variables in the analysis [60,61,62,63,64]. In terms of gender, a male student had a value of 1 and a female student 0. In terms of age, the 15-year-old group had the value of 1 and the 10-year-old group 0. In terms of the number of children, if the student was an only child, the value of 1 was assigned, otherwise 0. In terms of preschool education, if the student had preschool education experience, a value of 1 was assigned, otherwise 0. In terms of health, if the student was physically healthy, a value of 1 was assigned, otherwise 0.

The descriptive statistics of this study are shown in Table 1.

### 3.3. Methods and Models

#### 3.3.1. Baseline Regression

To examine the impacts of competitive school climate and cooperative school climate on the social and emotional skills of underprivileged-background students, this study used least squares regression, as shown in Equation (1)
(1)$${\mathrm{Social}\;\mathrm{and}\;\mathrm{Emotional}\;\mathrm{Skills}}_{i}=\beta_{0}+\beta_{1}\mathrm{coo}+\beta_{2}\mathrm{com}+\beta_{i}X_{i}+\varepsilon_{i}\;$$
where ${\mathrm{Social}\;\mathrm{and}\;\mathrm{Emotional}\;\mathrm{Skills}}_{i}$ denotes the scores of various domains of students’ social and emotional skills, coo is the score of the cooperative school climate, com is the score of the competitive school climate, $\beta_{1}$ is the regression coefficient of a cooperative school climate, $\beta_{2}$ is the regression coefficient of a competitive school climate, $\beta_{i}$ denotes the regression coefficients of the control variables, $X_{i}$ denotes students’ scores on the control variables, and $\varepsilon_{i}$ is the random error term.

#### 3.3.2. Quantile Regression

Since data samples are often not uniformly distributed, their regression coefficients are susceptible to the influence of extreme values [16]. In view of this, this study further adopted the quantile regression method to examine whether there was heterogeneity in the impacts of cooperative school climate and competitive school climate on underprivileged-background students with different levels of social and emotional skills, and Equation (2) is the model used.
(2)$$Q^{p}\left({\mathrm{Social}\;\mathrm{and}\;\mathrm{Emotional}\;\mathrm{Skills}}_{i}\right)=\beta_{0}^{p}+\beta_{1}^{p}\mathrm{coo}+\beta_{2}^{p}\mathrm{com}+\beta_{i}^{p}X_{i}+\varepsilon_{i}\;$$
where p is the quantile. In this study, the coefficients of dependent variables at different quantiles were calculated separately in a reversed order, including students in the bottom 10% quantile (with very low social and emotional skills), in the 25% quantile (with low social and emotional skills), in the 50% quantile (with moderate social and emotional skills), in the 75% quantile (with high social and emotional skills), and in the 90% quantile (with very high social and emotional skills).

## 4. Analysis

### 4.1. Baseline Regression

Stata 15.0 was used in the analysis. There were no serious collinearity problems and Heteroskedasticity-Robust + Standard + Errors in this study. The results are shown in Table 2. This study found that, with the relevant variables controlled for, a cooperative school climate had a significant positive impact on all five domains of underprivileged-background students: COL (β = 55.202, *p* < 0.01), EMO (β = 42.410, *p* < 0.01), ENG (β = 35.581, *p* < 0.01), OPE (β = 41.783, *p* < 0.01), and TAS (β = 40.701, *p* < 0.01); while a competitive school climate had a significant negative impact on the COL (β = −8.182, *p* < 0.01) and EMO (β = −6.757, *p* < 0.05) of underprivileged-background students. The empirical results also showed that male students only performed significantly better than female students in the domain of EMO (β = 14.990, *p* < 0.01); the 10-year-old group performed better than the 15-year-old group in all domains; students from families with more than one child performed better than those who were the only child at home in all domains; and healthy students only performed better than physically impaired students in the domain of ENG (β = 9.530, *p* < 0.05).

### 4.2. Quantile Regression

As multiple linear regression is the result of comparisons at the level of the sample mean and cannot fully take into account the heterogeneity in the sample, this study drew on Li’s study (2021) [16] and conducted quantile regressions on students with social and emotional skills at different quantile levels, to investigate the impact of sample heterogeneity on the regression results. According to the results of the quantile regressions shown in Table 3, the signs and significance of the coefficients of the impacts of a cooperative school climate and competitive school climate on the different domains of social and emotional skills of underprivileged-background students were broadly consistent with the results of the multiple linear regression model shown in Table 2.

This study also found that the quantile regression results showed different characteristics in different quantiles of different domains of social and emotional skills. Specifically, the regression coefficients of a cooperative school climate in all domains of social and emotional skills showed increasing trends. In the domain of COL, from the 10% quantile to the 90% quantile, the coefficient of the impact of a cooperative school climate grew from 43.328 (*p* < 0.01) to 64.368 (*p* < 0.01) when the cooperative school climate increased by one unit. In the domain of EMO, from the 10% quantile to the 90% quantile, the coefficient of the impact of a cooperative school climate grew from 36.593 (*p* < 0.01) to 49.955 (*p* < 0.01) when the cooperative school climate increased by one unit. In the domain ENG, from the 10% quantile to the 90% quantile, the coefficient of the impact of cooperative school climate grew from 26.501 (*p* < 0.01) to 45.347 (*p* < 0.01) when the cooperative school climate increased by one unit. In the domain of OPE, from the 10% quantile to the 90% quantile, the coefficient of the impact of a cooperative school climate grew from 29.243 (*p* < 0.01) to 49.224 (*p* < 0.01) when the cooperative school climate increased by one unit. In the domain of TAS, from the 10% quantile to the 90% quantile, the coefficient of the impact of a cooperative school climate grew from 29.835 (*p* < 0.01) to 46.142 (*p* < 0.01) when the cooperative school climate increased by one unit. In general, as social and emotional skills increased, cooperative school climates promoted the social and emotional skills of underprivileged-background students to an increasing extent. The 0.90 quantile saw the largest marginal effects of 64.368, 49.955, 45.347, 49.224, and 46.142, respectively, indicating a Matthew effect on the impacts of a cooperative school climate on underprivileged-background students, but the impacts of a cooperative school climate on the different social and emotional skills of underprivileged-background students showed significant homogeneity.

In contrast, the impacts of a competitive school climate on the social and emotional skills of underprivileged-background students in different quantiles showed pronounced heterogeneity. Specifically, as the social and emotional skills increased, the negative impacts of a competitive school climate on underprivileged-background students became weaker. In the domain of COL, from the 10% quantile to the 90% quantile, the coefficient of the impact of a competitive school climate changed from −16.940 (*p* < 0.01) to 2.190 (*p* > 0.1) when the competitive school climate increased by one unit. In the domain of EMO, from the 10% quantile to the 90% quantile, the coefficient of the impact of a competitive school climate changed from −15.601 (*p* < 0.01) to 4.518 (*p* > 0.1) when the competitive school climate increased by one unit. In the domain of ENG, from the 10% quantile to the 90% quantile, the coefficient of the impact of competitive school climate changed from −2.809 (*p* > 0.1) to 4.734 (*p* > 0.1) when the competitive school climate increased by one unit. In the domain of OPE, from the 10% quantile to the 90% quantile, the coefficient of the impact of a competitive school climate changed from −2.740 (*p* > 0.1) to 8.553 (*p* > 0.1) when the competitive school climate increased by one unit. In the domain of TAS, from the 10% quantile to the 90% quantile, the coefficient of the impact of a competitive school climate changed from −8.258 (*p* < 0.01) to 8.470 (*p* > 0.1) when the competitive school climate increased by one unit. These results suggested that as social and emotional skills increased, the negative impacts of a competitive school climate on underprivileged-background students gradually decreased and even shifted to insignificant positive impacts.

## 5. Conclusions and Discussion

Exploring the high-level development of underprivileged-background students has become an integral part of research on educational equity. However, there is a lack of studies on the impacts of school climate on various dimensions of the social and emotional skills of underprivileged-background students. Using the data from OECD-SSES Suzhou (China), this study assessed the mean effects of the impacts of cooperative school climate and competitive school climate on various dimensions of the social and emotional skills of underprivileged-background students and used the quantile regression method to examine the impact of sample heterogeneity on the regression results. On this basis, the findings are as follows: A cooperative school climate can improve various dimensions of the social and emotional skills of underprivileged-background students. A competitive school climate has limited negative impacts on the social and emotional skills of underprivileged-background students in China. The impacts of a cooperative school climate on underprivileged-background students with different levels of social and emotional skills were homogeneous. Meanwhile, the impacts of a competitive school climate on underprivileged-background students with different levels of social and emotional skills were heterogeneous. Specifically, the multiple linear regression results showed that, with the characteristics of the students and their families controlled for, cooperative school climates significantly promoted the major five domains of the social and emotional skills of underprivileged-background students (i.e., collaboration, emotion regulation, engagement with others, open-mindedness, and task performance), with marginal effects of 55.202, 42.410, 35.581, 41.783, and 40.701, respectively. Competitive school climates had significant negative impacts on underprivileged-background students in the domains of collaboration and emotional regulation, with marginal effects of −8.182 and −6.757, respectively, indicating that competitive climates did not have positive impacts on students in general. This study further explored the heterogeneity in the impacts of competitive school climate and competitive school climate on underprivileged-background students with different levels of social and emotional skills through conditional quantile regression. It was found that the impacts of cooperative school climate on the social and emotional skills of underprivileged-background students showed homogeneity and significantly promoted various dimensions in different quantiles, indicating that school cooperative climates have significant positive impacts on students with different levels of social and emotional skills. In contrast, the impacts of a competitive school climate on the social and emotional skills of underprivileged-background students showed significant heterogeneity, with the influence increasing from a low level to a high level as the quantile moved upward. As social and emotional skills increased, the negative impacts of a competitive school climate on underprivileged-background students weakened and even shifted to insignificant positive impacts.

In China, studies on underprivileged-background students have mainly been conducted from the perspective of material support and academic achievement, and there is a lack of large-scale empirical studies investigating the development of the social and emotional skills of underprivileged-background students, from the perspective of cooperative school climate and competitive school climate. An essay titled *Record on the Subject of Education* in a Chinese classic 2000 years ago argued that it is difficult for one studying alone to become knowledgeable, emphasizing the importance of cooperation and communication with peers in the learning process, to draw on each other’s strengths and make progress together. This study controlled for relevant variables and found that underprivileged-background students had better development of social and emotional skills in more cooperative school climates, which is consistent with the findings of some previous studies [57,65,66]. Most of the current studies on cooperative school climate have been carried out from the perspective of cooperative learning. Cooperative learning underscores social interaction among students within learning groups and the ability to work together to solve problems [67]. In such a context, students are required to build learning communities based on the concept of cooperative learning with familiar and unfamiliar peers, while participating in cognitive restructuring through interaction, accepting divergent ideas, and developing previously imperfect mental models [16,68,69]. At the same time, pressure-induced negative impacts can be alleviated through the social and emotional support obtained from cooperation with other students and teachers [70]. Cooperative activities create more opportunities to interact with others and prevent underprivileged students from overly suppressing their emotions, which can cause emotional disorders when they have individual dilemmas and negative interpersonal relationships, which is why those who engage in cooperation tend to show more positive emotions and pro-social behavior than those who do not [2,71,72,73]. In addition, Slavin (1991) pointed out that successful cooperative learning has two key elements: clear learning goals and personal responsibility [74]. During cooperative learning, scores and rankings of teams are closely related to the collaboration among team members. Cooperative members show higher levels of task motivation, to help them do better and promote students’ personal responsibility, which helps boost their task performance [75,76,77]. Therefore, cooperative school climates can promote the development of various dimensions of the social and emotional skills of underprivileged-background students.

This study also observed that the relationship between the preschool education experience and the development of the social-emotional skills of underprivileged students is highly complex, such that there is no uniform promoting effect or inhibiting effect at work. Moreover, the analysis results showed that preschool education experience only had a significant positive effect on EMO in the students from underprivileged backgrounds (β = 20.173 *p* < 0.1), whilst the other skill dimensions had no significant effect. Such a finding is notable, as it is in contrast to the observations published in some of the existing literature on this topic. In previous studies, researchers generally concluded that preschool education experience can promote the development of students’ social-emotional skills [63,64,78]. This study argues that there are two main reasons for this: First, previous studies were mostly conducted on samples consisting of lower-grade students in primary school [78] and the measured results were significantly influenced by their experiences in preschool education. In contrast, this study mainly used students aged 10 and 15 years old, meaning that the effect of preschool education experience on students at this stage is likely to be eliminated or obscured by other influencing factors, such as the quality of primary or secondary education [79,80]. Second, previous studies often use regression technology to analyze the data of all student samples [63,64], meaning that students from underprivileged and other backgrounds were included. Due to the heterogeneity of the social and emotional skills of students from underprivileged and other backgrounds, research conclusions are susceptible to extreme values. Accordingly, the measured average influence of preschool experience is easily influenced by the presence of advantaged students in the sample.

## 6. Implications and Research Prospects

Schools can create a favorable cooperative school climate from the following perspectives. First, school leaders should ensure that teachers can carry out cooperative teaching through implementing policies, establish diverse cooperative platforms for underprivileged-background students, and create more opportunities for cooperation in other contexts, such as after-school activities and community services, instead of being confined to the classroom. Second, teachers should reasonably arrange the mode and time of cooperation when carrying out cooperative learning and appropriately divide students of different levels and specialties into different groups, to develop personal responsibility, interdependence, and effective interaction, and to prevent students from slacking off or shirking their responsibilities [81,82]. Third, teachers, whether in classroom cooperative learning or extracurricular cooperative activities, should encourage students with different strengths to form learning communities, thus enhancing their strengths and making up for weaknesses through cooperation.

This study, focused on the impacts of school climate on the development of the social and emotional skills of underprivileged-background students, conducted an empirical analysis based on large-scale survey data. This makes up for the shortcomings of the existing studies to some extent, enriches the research on the development of the social and emotional skills of specific groups, and provides scientific evidence for improving the school climate and management in primary and secondary schools. However, this study has some limitations that need to be considered in the future. First, limited by the data selected, this study could not dig deeper into the mechanism connecting school climate and students’ social and emotional skills at the empirical level. Second, the simple division of cooperation and competition into two dimensions may result in a loss of information. Future studies could be conducted based on the present study, to investigate the impacts of different combinations of the various types of school climate on adolescents’ social and emotional skills. Third, this study found that the marginal effect of preschool education experience on student development may be different for students from different socioeconomic backgrounds. With this in mind, a follow-up study could further explore the heterogeneity of the effect of preschool education experience on the development of students from different socioeconomic backgrounds. Fourth, as the data used in this study were mainly derived from a survey of Chinese students, the conclusions obtained may be mediated by the social, economic, and cultural background of China. Hence, it follows that the conclusions obtained may not be directly applicable to other countries. Future studies could analyze the survey data of different countries, to compare their respective differences regarding the influence of cooperative and competitive school climates on students’ development for different countries and different social and cultural backgrounds.

## Figures and Tables

**Table 1 children-09-01812-t001:** Descriptive statistics.

Variables	Mean	SD	Min	Max
COL	614.82	88.28	388.55	901.28
EMO	534.92	82.16	215.14	927.85
ENG	544.33	67.26	281.31	888.33
OPE	588.24	77.23	394.66	911.84
TAS	585.52	80.41	322.43	882.74
Cooperative School Climate	2.64	0.72	1	4
Competitive School Climate	2.24	0.79	1	4
Gender	0.56	0.50	0	1
Age	0.52	0.50	0	1
Only Child	0.37	0.48	0	1
Preschool Education Experience	0.99	0.12	0	1
Physical Health	0.90	0.30	0	1

**Table 2 children-09-01812-t002:** Results of the path analysis model.

Variables	COL (N = 1739)	EMO (N = 1739)	ENG (N = 1739)	OPE (N = 1739)	TAS (N = 1739)
Cooperative School Climate	55.202 ***	42.410 ***	35.581 ***	41.783 ***	40.701 ***
	(2.77)	(2.81)	(2.33)	(2.73)	(2.77)
Competitive School Climate	−8.182 ***	−6.757 **	0.591	1.771	−0.124
	(2.79)	(2.96)	(2.22)	(2.66)	(2.78)
Gender	−2.395	14.990 ***	1.804	−1.977	−0.752
	(3.55)	(3.53)	(2.83)	(3.30)	(3.45)
Age	−56.376 ***	−42.640 ***	−39.606 ***	−40.661 ***	−42.417 ***
	(3.85)	(3.79)	(3.03)	(3.55)	(3.72)
Only Child	−9.543 ***	−9.557 ***	−10.094 ***	−8.174 **	−14.083 ***
	(3.62)	(3.60)	(2.91)	(3.33)	(3.49)
Preschool Education Experience	13.459	20.173 *	−8.832	−5.283	10.198
	(11.09)	(11.05)	(10.04)	(11.45)	(3.76)
Physical Health	4.811	8.822	9.530**	6.519	8.938
	(5.86)	(5.49)	(4.72)	(5.00)	(5.88)
Cons.	503.953 ***	427.549 ***	472.544 ***	498.589 ***	488.171 ***
	(15.02)	(14.57)	(13.09)	(14.42)	(16.33)
R2	0.337	0.240	0.256	0.239	0.228
Adj.R2	0.334	0.237	0.253	0.236	0.225
F	113.646 ***	65.031 ***	67.392 ***	60.098 ***	61.923 ***

Note: (1) Standard errors are in parentheses; (2) *** = *p* < 0.01, ** = *p* < 0.05, * = *p* < 0.1.

**Table 3 children-09-01812-t003:** Results of the quantile regression analysis model.

	Q10	Q25	Q50	Q75	Q90
COL (N = 1739)					
Cooperative School Climate	43.328 ***	49.039 ***	53.798 ***	59.391 ***	64.368 ***
	(3.01)	(2.98)	(2.60)	(3.42)	(5.58)
Competitive School Climate	−16.940 ***	−11.364 ***	−11.646 ***	−9.302 ***	2.190
	(2.85)	(2.83)	(2.47)	(3.25)	(5.30)
Control variables	YES	YES	YES	YES	YES
Cons.	456.492 ***	472.277 ***	521.118 ***	554.000 ***	539.819 ***
	(21.91)	(21.71)	(18.93)	(24.94)	(40.68)
P.R^2^	0.119	0.158	0.189	0.245	0.268
EMO (N = 1739)					
Cooperative School Climate	36.593 ***	32.913 ***	38.838 ***	41.538 ***	49.955 ***
	(4.32)	(2.37)	(2.18)	(3.19)	(5.88)
Competitive School Climate	−15.601 ***	−10.155 ***	−6.942 ***	−7.778 **	4.518
	(4.11)	(2.25)	(2.07)	(3.03)	(5.58)
Control variables	YES	YES	YES	YES	YES
Cons.	382.240 ***	410.204 ***	430.356 ***	469.554 ***	495.761 ***
	(31.52)	(17.28)	(15.87)	(23.23)	(42.86)
P.R^2^	0.093	0.111	0.130	0.152	0.182
ENG (N = 1739)					
Cooperative School Climate	26.501 ***	26.647 ***	32.077 ***	40.371 ***	45.347 ***
	(2.75)	(2.1)	(2.09)	(2.95)	(4.46)
Competitive School Climate	−2.809	−0.038	−2.736	3.193	4.734
	(2.62)	(1.99)	(1.99)	(2.80)	(4.24)
Control variables	YES	YES	YES	YES	YES
Cons.	441.071 ***	443.980 ***	474.977 ***	490.073 ***	513.286 ***
	(20.08)	(15.28)	(15.25)	(21.48)	(32.52)
P.R^2^	0.099	0.111	0.133	0.159	0.194
OPE (N = 1739)					
Cooperative School Climate	29.243 ***	33.331 ***	37.966 ***	42.747 ***	49.224 ***
	(2.97)	(2.46)	(2.45)	(3.41)	(5.76)
Competitive School Climate	−2.740	−4.128 *	0.666	0.983	8.553
	(2.82)	(2.34)	(2.32)	(3.24)	(5.47)
Control variables	YES	YES	YES	YES	YES
Cons.	468.660 ***	480.387 ***	506.900 ***	536.682 ***	536.996 ***
	(21.64)	(17.94)	(17.84)	(24.89)	(41.99)
P.R^2^	0.074	0.092	0.118	0.155	0.195
TAS (N = 1739)					
Cooperative School Climate	29.835 ***	34.722 ***	42.365***	46.813 ***	46.142 ***
	(2.76)	(2.5)	(2.81)	(3.76)	(5.42)
tCompetitive School Climate	−8.258 ***	−4.247 *	0.097	1.397	8.470
	(2.62)	(2.37)	(2.66)	(3.57)	(5.15)
Control variables	YES	YES	YES	YES	YES
Cons.	439.532 ***	467.167 ***	466.241 ***	521.143 ***	543.259 ***
	(20.09)	(18.19)	(20.45)	(27.42)	(39.52)
P.R^2^	0.071	0.087	0.111	0.166	0.209

Note: (1) Standard errors are in parentheses; (2) *** = *p* < 0.01, ** = *p* < 0.05, * = *p* < 0.1; (3) P.R^2^= Pseudo R^2^; (4) Yes means that the variable of interest has been controlled for in the quantile regression.

## Data Availability

The data are available online at https://www.oecd.org/education/ceri/social-emotional-skills-study/ (accessed on 15 September 2021).

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
