# Peer review of "How School Climate Affects the Development of the Social and Emotional Skills of Underprivileged-Background Students—An Empirical Study Based on the SSES2019 Data"

_children, 2022, doi:10.3390/children9121812_

Round 1

Reviewer 1 Report

Dear authors, 

Congratulations on your research which has been explained in a very appropriate way, including a good introduction, good data analysis, nice tables, and very good conclusions.

However, the abstract should be modified to follow the APA instructions. The abstract should have two lines of introduction framing the research, which is almost done, followed by the sample, the instruments, results, and conclusions.

Additionally those results with a lack of correlations, such as the preschool experience with the dependent variable, it would deserve some explanations.

Thanks for providing me with the opportunity to review this paper.

Author Response

Dear editors and dear reviewers,

Thank you for your letter and the reviewers’ comments concerning our manuscript entitled "How School Climates Affect the Development of the Social and Emotional Skills of Underprivileged Background Students—An Empirical Study Based on the SSES2019 Data" (ID: children-1994891).

Those comments are valuable and very helpful. We have read through the comments carefully and have made corrections. Based on the instructions provided in your letter, we uploaded the file of the revised manuscript. Revisions in the text are shown using the “track changes” function for additions, and strike through font for deletions. The responses to the reviewer's comments are marked in red and presented following.

The main corrections in the paper and the responds to the reviewer’s comments are as flowing:

Response to reviewer #1:

Q1

The abstract should be modified to follow the APA instructions. The abstract should have two lines of introduction framing the research, which is almost done, followed by the sample, the instruments, results, and conclusions.

Response

Thank you for your suggestions pertaining to how we can go about optimizing the abstract portion of this manuscript. We are of the mind that it is both necessary and reasonable to optimize the abstract. In response to your suggestions, we have revised the abstract so it is in accordance with the APA instructions. The revised content is as follows (alternatively, please refer to the Revised Manuscript Abstract section).

Modified abstract reads

Abstract:

Background: Promoting the development of the social and emotional skills of underprivileged background students is an integral part of educational and social equity. To date, there is a lack of relevant research in this field.

Aims: This study investigated the impacts of cooperative school climates and competitive school climates on the development of the social and emotional skills of underprivileged background students.

Sample: This study used the data of Chinese underprivileged background students(N=1739) from the Study on Social and Emotional Skills conducted by the Organisation for Economic Cooperation and Development (OECD-SSES2019).

Methods: This study selected the ordinary least squares (OLS) analysis method and the quantile regression (QR) analysis method.

Results: The ordinary least squares (OLS) analysis results showed that cooperative school climates promoted the development of various dimensions of social and emotional skills of underprivileged background students, while competitive school climates had significant negative impacts on collaboration and emotional regulation of underprivileged background students and had no significant impact on the other three major domains, namely engagement with others, open-mindedness, and task performance. The quantile regression analysis further explored the heterogeneity in the impacts of cooperative school climate and competitive school climate on the development of the social and emotional skills of underprivileged background students through quantile regressions and found that the impacts of competitive school climates on underprivileged background students with different levels of social and emotional skills are homogeneous, while the impacts of competitive school climates on underprivileged background students with different levels of social and emotional skills were heterogeneous.

Conclusions: These findings provide greater insight into the roles of the school cooperative climate and the school competitive climate in the development process of the social and emotional skills of underprivileged background students.

Q2:

Additionally those results with a lack of correlations, such as the preschool experience with the dependent variable, it would deserve some explanations.

Response:

Thank you for putting forward potential areas for improvement in the discussion section of this manuscript. Following your comments, we have incorporated a discussion of the influence of the preschool education experience on the social and emotional skills’ development of students from underprivileged backgrounds. We also adopt the view that the influence of preschool education experience on the development of students' social and emotional skills may exhibit some heterogeneity in relation to the family socio-economic background or age dimensions. However, as this topic is not the focus of the present manuscript, suggestions for improvement in future research were added to the third part of the manuscript (“Research Prospect”). The specific supplementary contents are as follows. The corresponding content marked in red in "Conclusion and Discussion" and "Implications and Research Prospect" can also be found in the revised manuscript.

Additional content in the Conclusion and Discussion

This study also observed that the relationship between preschool education experience and the development of the social-emotional skills of underprivileged students is highly complex, such that there is no uniform promoting effect or inhibiting effect at work. Moreover, the analysis results showed that preschool education experience only had a significant positive effect on EMO in the students from underprivileged backgrounds (β=20.173 p<0.1), whilst the other skill dimensions had no significant effect. Such a finding is notable as it is in contrast to the observations published in some of the existing literature on this topic. In previous studies, researchers generally concluded that preschool education experience can promote the development of students’ social-emotional skills [63-64][78]. This study argues that there are two main reasons for this: First, previous studies were mostly conducted on the samples consisting of lower-grade students in primary school [78] and the measured results were significantly influenced by their experiences in preschool education. In contrast, this study mainly uses students aged 10 and 15 years old, meaning that the effect of preschool education experience on students at this stage is likely to be eliminated or obscured by other influencing factors, such as the quality of primary or secondary education [79-80]. Second, previous studies often use regression technology to analyze the data of all student samples [63-64], meaning that students from underprivileged and other backgrounds are included. Due to the heterogeneity of the social and emotional skills of students from underprivileged and other backgrounds, the research conclusions are susceptible to extreme values. Accordingly, the measured average influence effect of preschool experience is easily influenced by the presence of advantaged students in the sample.

Additional content in the Implications and Research Prospect

Thirdly, this study finds that the marginal effect of preschool education experience on student development may be different for students from different socioeconomic backgrounds. With this in mind, the follow-up study can further explore the heterogeneity of the effect of the preschool education experience on the development of students from different socioeconomic backgrounds.

Response to reviewer #2:

Q1

Introduction: The introduction and need for the research study is clear. I suggest a more detailed description of what the authors mean by equality vs equity. For example, the authors used "equality" in line 43, but it seems they might want to mean equity.

Response

We apologize for the language problems in the original manuscript. In line 43, we are indeed referring to "equity" and have made revisions at the corresponding position for you to review. Although a discussion of the relationship between "equality" and “equity s" is beyond the scope of this study, we believe that it is nevertheless necessary to introduce the definition and comparison of "equality" and "equity" here, per your suggestion. Equity and equality are the core goals of social development and have also been widely discussed as central issues in political philosophy, moral philosophy, and social science (Yang, 2004). The present study regards "equality" as a relatively restrictive concept referring to the distribution of basic personality and rights, important resources, and necessary abilities, whereas "equity” is a relatively open concept that is more applicable to the distribution of social resources and public rights sit. lt can be said that "equality" is the lowest level and the most primitive form of "equity" (Yu, 2017; Ran, 2008). As both a subsystem and an important basis of the social equity system, educational equity not only embodies the values of equity and justice in the educational system, but is also the basic value of educational modernization and the starting point of various countries’ educational policies (Chu, 2006; Feng, Gao, 2022; Guo, 2000; Yang, 2000). Equality is one of the core values of educational equity, which is inseparable from both the promotion of educational equality and the improvement of educational efficiency (Si, Song, 2008; Ran, 2008).

Reference:

Yang, L. On comparing equality with fairness, justice, and impartiality. Journal of Literature, History & Philosophy 2004, 04, 145-151.

Yu, K. Rethinking equality, fairness, and justice. Academic Monthly 2017, 49(04), 5-14.

Ran, Y. Equality and efficiency: the core values of educational equity. Journal of Teaching and Management 2008, 15, 3-5.

Feng, J.; Gao, Z. Policy orientation and practice exploration of educational equity in the new era of China. Journal of Northeast Normal University (Philosophy and Social Sciences) 2022, 04, 16-23.

Guo,Y. Theoretical reflections on the issue of equity in education. Educational Research 2000, 03, 31-34+47.

Yang, D. Awareness and reflection on the issue of equity in education in China. Research in Educational Development 2000, 08, 5-8.

Chu, H. Some basic theoretical questions about equity in education. Journal of The Chinese Society of Education 2006, 12, 1-4.

Si, X.; Song, H. Educational justice is the ideal state of the harmonious development of equal education and educational efficiency. Theory and Practice of Education 2008, 16, 27-30.

Q2:

The authors recommendations on cooperative classrooms are commendable. I suggest the authors expand on cultural implications and limitations. Given that the study was conducted in China, one could infer that culture, language and government policies might greatly differ from other countries (particularly, Western cultures). A discussion on culture seems appropriate here. Also, this might place a limitation on the generalizability of the results and conclusions.

Response

We appreciate the reviewer’s positive evaluation of our work and agree with the comments regarding the limitations of our study. As you have rightly noted, the development of student’s social and emotional skills is influenced by regional social, economic, cultural, and educational policies. On this basis, it is reasonable to limit the research sample to one country, as doing so will allow us to articulate more specific suggestions for students' development in light of the country or region’s characteristics. At the same time, we also believe that the comparative study between multiple countries and multiple cultures is also necessary, such that future research should conduct further analysis on this basis. Following the discussion by the research group, we have responded to and supplemented your suggestions in the fourth part of "Research Prospect" in this manuscript. The specific reply content is as follows (alternatively refer to the "Implications and Research Prospect" section of the revised draft).

Additional content in the Implications and Research Prospect

Fourth, as the data used in this study is mainly derived from a survey of Chinese students, the conclusions obtained may be mediated by the social economic, and cultural background of China. Hence, it follows that the conclusions obtained may not be directly applied to other countries. Future studies can analyze the survey data of different countries to compare the respective differences in the influence of cooperative and competitive school climate on students’ development between different countries and different social and cultural backgrounds.

Response to reviewer #3:

Thank you for your suggestions on the concept of "school climate" in this manuscript. We reviewed articles about the positive school climates(see Berkowitz, R. School Matters: The Contribution of Positive School Climate to Equal Educational Opportunities among Ethnocultural Minority Students. Youth & Society 2020, 54, 372 - 396.; Zullig, K.J.; Koopman, T.M.; Patton, J.M.; Ubbes, V.A. School Climate: Historical Review, Instrument Development, and School Assessment. Journal of Psychoeducational Assessment 2010, 28, 139 - 152.; Cornell, D. Research summary for the authoritative school climate survey. Dewey Cornell 2019.) and the technical documents and questionnaires published by the OECD. After discussion by the research group, we replied to the review comments as follows.

Q1

I found this article pleasant to read. It explained how the study was carried out in very good detail. However, there are some issues that the editor will list, which issues I feel preclude me from suggesting immediate publication. My one major problem is that the notion of school climate is not dealt with closely enough and that you do not describe what a school with a good school climate is. There are specific criteria for a positive school climate (see La Salle, T. P., 2017. Technical manual for the Georgia School Climate Survey Suite. Georgia Department of Education). I fear there is a conflation of school climate and cooperative learning. Indeed, the latter is only one aspect of the former.

Response

Using data from the OECD-SSES survey, this study focuses on the respective influence and heterogeneity of cooperative and competitive school atmospheres on the social and emotional competence of students from disadvantaged backgrounds. "在你所在的学校中,同学们看起来重视合作(如:一起学习)About your school, students seem to value cooperation (e.g. working together" and "在你所在的学校中,看起来同学们在相互合作。About your school, it seems that students are cooperating with each other." represent "cooperative school climate" whilst questions on "competitive school climate" are represented by "在你所在的学校中,同学们看起来重视竞争(如:相互竞争)。About your school, students seem to value competition (e.g. competing with each other)." and "在你所在的学校中,看起来同学们都在相互竞争。About your school, it seems that students are competing with each other.". In addition, the technical document of the survey classifies these four questions as "School Climate". Therefore, the issues investigated in this study are not strongly related to "cooperative learning", but rather the survey questions of "the cooperative school climate " and " the competitive school climate" as defined for present purposes (note: the technical documentation and questionnaire are available at https://www.oecd.org/education/ceri/social-emotional-skills-study/data.htm). In addition, the dimensions and questions included in the OECD survey have passed strict expert review, semantic analysis, and reliability and validity testing. We believe that the definitions of "cooperative school climate" and "competitive school climate" presented in the OECD questionnaire and technical documents are reasonable and applicable to this study. To sum up, we posit that the items used in the core independent variables in this paper generally point to the concepts of "cooperative school climate" and "competitive school climate", not other concepts, such as "cooperative learning” and "positive school climate".

Q2:

It would not go amiss if you had to delve more deeply into the details of positive school climates as background material for this study.

Response

We have briefly introduced the definition of "positive school climate" in this study in footnote P3 of the original manuscript. However, as the focus of this manuscript does not include a discussion of the impact of a positive school climate on students' development, the text does not provide much by way of an introduction to the “positive school climate” concept. Even so, we still believe that the "positive school climate" concept is an important area of study. It is readily apparent from reviewing the existing research that a good school atmosphere can promote the improvement of students’ academic performance and the development of their social and emotional skills, whilst at the same time minimizing the negative impact of disadvantaged family backgrounds on students' development. Unfortunately, the OECD-SSES (2019) did not include a question specifically measuring "positive school climate”, though this will be something our team will focus on in future research.

The contents above are the authors' responses to the reviewers' comments, and I would be grateful if the editorial board and reviewers could review them.

In addition, we are grateful to the editors and reviewers for pointing out corrections to possible shortcomings in this manuscript and for providing directions in which improvements can be made. It is a wonderful opportunity for us to revisit possible problems in this manuscript and has led to its total enhancement and deepening!

We would love to thank you for allowing us to resubmit a revised copy of the manuscript and we highly appreciate your time and consideration.

Best regards,

Jijun Yao,

Nov. 15, 2022

Reviewer 2 Report

Thank you for the opportunity to review your manuscript titled " How School Climates Affect the Development of the Social and Emotional Skills of Underprivileged Background Students— An Empirical Study Based on the SSES2019 Data" The study is an important contribution to the research base. I suggest the following to make it stronger.

Introduction: The introduction and need for the research study is clear. I suggest a more detailed description of what the authors mean by equality vs equity. For example, the authors used "equality" in line 43, but it seems they might want to mean equity.

Methods and results: These are robust and well-explained. Also provide an important contribution to the research base.

Conclusions and implications, and limitations: The authors recommendations on cooperative classrooms are commendable. I suggest the authors expand on cultural implications and limitations. Given that the study was conducted in China, one could infer that culture, language and government policies might greatly differ from other countries (particularly, Western cultures). A discussion on culture seems appropriate here. Also, this might place a limitation on the generalizability of the results and conclusions.

Author Response

(The authors gave the same response as above.)

Reviewer 3 Report

I found this article pleasant to read. It explained how the study was carried out in very good detail. However, there are some issues that the editor will list, which issues I feel preclude me from suggesting immediate publication. My one major problem is that the notion of school climate is not dealt with closely enough and that you do not describe what a school with a good school climate is. There are specific criteria for a positive school climate (see La Salle, T. P., 2017. Technical manual for the Georgia School Climate Survey Suite. Georgia Department of Education). I fear there is a conflation of school climate and cooperative learning. Indeed, the latter is only one aspect of the former.

Apart from that, some incomplete sentences need to be checked, and the whole article could benefit from some pruning or shortening. It would not go amiss if you had to delve more deeply into the details of positive school climates as background material for this study.

I hope that your article will be published soon.

Author Response

(The authors gave the same response as above.)
